# Evaluating Rabies Test Accuracy: A Systematic Review and Meta-Analysis of Human and Canine Diagnostic Methods

**DOI:** 10.3390/diagnostics15040412

**Published:** 2025-02-08

**Authors:** Mayron Antonio Candia-Puma, Leydi Pola-Romero, Haruna Luz Barazorda-Ccahuana, Luis Daniel Goyzueta-Mamani, Alexsandro Sobreira Galdino, Ricardo Andrez Machado-de-Ávila, Rodolfo Cordeiro Giunchetti, Eduardo Antonio Ferraz Coelho, Miguel Angel Chávez-Fumagalli

**Affiliations:** 1Computational Biology and Chemistry Research Group, Vicerrectorado de Investigación, Universidad Católica de Santa María, Arequipa 04000, Peru; mcandia@ucsm.edu.pe (M.A.C.-P.); hbarazorda@ucsm.edu.pe (H.L.B.-C.); lgoyzueta@ucsm.edu.pe (L.D.G.-M.); 2Facultad de Ciencias Farmacéuticas, Bioquímicas y Biotecnológicas, Universidad Católica de Santa María, Arequipa 04000, Peru; 3Laboratório de Biotecnologia de Microrganismos, Universidade Federal São João Del-Rei, Divinópolis 35501-296, Brazil; asgaldino@ufsj.edu.br; 4Instituto Nacional de Ciência e Tecnologia em Biotecnologia Industrial, INCT-BI, Distrito Federal, Brasilia 70070-010, Brazil; 5Programa de Pós-Graduação em Ciências da Saúde, Universidade do Extremo Sul Catarinense, Criciúma 88806-000, Brazil; r_andrez@unesc.net; 6Laboratório de Biologia das Interações Celulares, Instituto de Ciências Biológicas, Universidade Federal de Minas Gerais, Belo Horizonte 31270-901, Brazil; giunchetti@icb.ufmg.br; 7Instituto Nacional de Ciência e Tecnologia em Doenças Tropicais, INCT-DT, Salvador 40015-970, Brazil; 8Infectologia e Medicina Tropical, Faculdade de Medicina, Universidade Federal de Minas Gerais, Belo Horizonte 31270-901, Brazil; eduardoferrazcoelho@yahoo.com.br

**Keywords:** rabies, diagnostic tests, meta-analysis, systematic review, sensitivity, specificity

## Abstract

**Background/Objectives:** Rabies is almost invariably fatal once clinical symptoms manifest. Timely and accurate diagnosis is essential for effective treatment and prevention. Dogs are the principal reservoirs of the virus, particularly in developing nations, highlighting the importance of precise diagnostic and control measures to prevent human cases. This systematic review and meta-analysis assessed the accuracy of laboratory tests for diagnosing rabies in humans and dogs. **Methods:** The PubMed database was searched for published studies on rabies diagnosis between 1990 and 2024. Following PRISMA statement recommendations, we included 60 studies that met the selection criteria. **Results:** The results demonstrated the effectiveness of immunological tests like the Enzyme-Linked Immunosorbent Assay (ELISA) and molecular tests such as Reverse Transcription Polymerase Chain Reaction (RT-PCR) for both humans and dogs. In this study, the Direct Fluorescent Antibody Test (DFAT) exhibited lower diagnostic performance, with an area under the curve for false positive rates (AUC_FPR_ = 0.887). In contrast, ELISA (AUC_FPR_ = 0.909) and RT-PCR (AUC_FPR_ = 0.905) provided more consistent results. Notably, the Rapid Immunochromatographic Test (RIT) showed the best performance (AUC_FPR_ = 0.949), highlighting its superior diagnostic capabilities compared to DFAT. **Conclusions:** These findings underscore the need to modernize rabies diagnostic protocols by incorporating advanced methodologies to improve diagnostic accuracy, reduce transmission, and decrease mortality rates.

## 1. Introduction

Rabies is an infectious disease caused by Lyssavirus genus members and remains a major public health burden worldwide [1,2]. The World Health Organization estimates that around 59,000 deaths result from rabies yearly and therefore emphasizes the need for effective control and prevention measures [3]. A high mortality rate is associated with rabies, almost 100% once clinical symptoms develop [4]. Rabies is nearly always fatal if post-exposure prophylaxis (PEP) is not administered promptly following exposure. PEP, which includes a series of rabies vaccinations and immunoglobulin therapy, is effective in preventing symptom onset if given before the virus invades the central nervous system [5,6]. Although preventive measures have advanced, no universally effective treatment exists for rabies once neurological symptoms develop [7]. Experimental therapies, like the Milwaukee protocol, have shown limited success, highlighting the critical need for early intervention and developing more effective therapeutic strategies [8]. This emphasizes the need for a quick and correct diagnosis to enable timely intervention [9]. Delays or errors in diagnosing rabies could lead to the loss of an opportunity to provide PEP and would contribute to ongoing rabies transmission [10].

Rabies is also one of the most concerning veterinary diseases [11]. Dogs are the major reservoir and transmitter of rabies to humans [12]. Most human rabies cases are associated with dog bites in most developing countries [13]. Because human rabies can be prevented, controlling rabies in dog populations can prevent almost all human cases [14]. Veterinary public health activities—particularly mass dog vaccination campaigns, population management strategies, and adequate dog population healthcare facilities—have become important components of many rabies control programs [14,15]. However, the effectiveness of these activities depends on the ability to diagnose rabies in animals [16]. In cases of misdiagnosis, inappropriate management of suspected cases, either through unnecessary culling or by failure to control an outbreak, will be realized and consequently affect animal welfare and public health [17]. It is crucial to highlight that dogs infected with rabies are seldom treated, due to the significant risk of transmission and the absence of effective treatment options for animals [18].

Additionally, rabies transmission via bats has been well-documented, presenting considerable challenges in urban and rural areas. As nocturnal and elusive carriers of the rabies virus, bats complicate control measures significantly [19]. Likewise, wild animals, such as raccoons, skunks, foxes, and coyotes, serve as substantial reservoirs of the virus in different regions [20]. This underscores the diverse transmission vectors of rabies and the complexities involved in its control, thereby needing comprehensive efforts encompassing both domestic and wild animals [21].

Diagnosis in humans and dogs must be accurate so that effective early intervention, proper treatment, and effective disease management may be carried out [9]. Among the numerous laboratory tests employed for diagnosing rabies are direct fluorescent antibody test (DFAT), polymerase chain reaction (PCR) assay, and various immunological assays [16,22]. The gold standard for post-mortem diagnosis has been the DFAT, detecting rabies virus antigens in brain tissue samples [23]. PCR will also detect the same viral RNA with extreme sensitivity from saliva, cerebrospinal fluid, and tissue samples, hence allowing antemortem diagnosis [24]. The serum samples are tested for rabies virus-specific antibodies in immunological assays, including the enzyme immunoassay; recently, the protein A and neutralizing peroxidase test have replaced many other tests [25,26,27]. Despite the availability of these, there is significant variability in the reported accuracy of these tests, which could affect clinical and public health outcomes. The performance of a test may depend on factors such as the stage of the infection, the quality of sample collection, and the specific protocols employed [28]. Additionally, advancements in diagnostic technology, such as next-generation sequencing (NGS) and novel biomarkers, offer the potential for enhancing the sensitivity and specificity of rabies diagnosis. However, these methods necessitate further validation and standardization before broad implementation in clinical settings [29,30].

This work aims to compile data on the overall diagnostic performance of laboratory tests for rabies in humans and dogs, as well as their sensitivity and specificity. In doing so, we hope to point out the most accurate rabies diagnostic instruments to aid in better clinical judgment and public health initiatives.

## 2. Materials and Methods

### 2.1. Study Protocol

This systematic review followed the guidelines set by the Preferred Reporting Items for Systematic Reviews and Meta-Analyses (PRISMA), which can be seen in Appendix A [31]. The review protocol was registered on the International Platform of Registered Systematic Review and Meta-analysis Protocols (INPLASY) website, under the registration number INPLASY2024110019. The complete protocol can be accessed at https://inplasy.com/inplasy-2024-11-0019/ (accessed on 4 November 2024).

### 2.2. Eligibility Criteria

This systematic review incorporates studies that evaluate the diagnostic accuracy of laboratory tests for rabies in humans and dogs by analyzing their sensitivity and specificity. We included randomized controlled trials, observational studies, and cohort studies published in peer-reviewed journals. Studies had to present enough data for the computation of diagnostic accuracy measures. Exclusion criteria included papers with no original data, reviews, case reports, editorials, and those not in English. Also, studies with major methodological flaws or incomplete data have been excluded from this review to ensure reliability and validity in their findings. A decision for the final selection of the studies was made following a careful screening of titles, abstracts, and full texts by two reviewers (M.A.C.-P. and L.P.-R.). Any disagreements were resolved by discussion or consulting a third reviewer (M.A.C.-F.).

### 2.3. Information Sources and Search Strategy

We utilized the MeSH terms “*Rabies*” and “*Laboratory Diagnosis*” to identify related terms for diagnosing rabies in the biomedical literature. Visualization was obtained by creating a network diagram of MeSH term co-occurrence using VOSviewer software (version 1.6.20) [32]. To provide more focus on searching for terms related to tests, we checked clusters within the network map. Subsequently, a second round of searches resulted from the combination of each MeSH term obtained in the cluster analysis with the MeSH terms “*sensitivity and specificity*”, which, meanwhile, are standard indicators for the evaluation of test performance in the clinical field [33], and “*rabies*”. Bibliographic studies were retrieved from the PubMed database (https://pubmed.ncbi.nlm.nih.gov/; last accessed 12 June 2024) between 1990 and 2024.

### 2.4. Study Selection and Data Collection Process

The review of studies had a selection process that was carried out through three key stages: identification, screening, and eligibility. We included all human and dog patients’ studies published from 1990 to 2024. Duplicate studies, non-English publications, review papers, and meta-analyses were among the exclusion criteria for this study. All the relevant titles and abstracts of identified studies were screened. In this phase of eligibility, full texts were selected and classified as highly relevant to the research question and thinned down according to the studies that had worked with diagnostic tests for rabies.

Data were extracted regarding the diagnostic test used in each study, the type of diagnostic test, the number of patients with rabies, the type of experimental subjects (human/dog), and the sample type. Traditional diagnosis methods, such as culture and histopathology, were not included in our review since we decided to focus on molecular and immunological testing protocols that have gained clinical and research evidence in early rabies infection detection. We only included studies that calculated some measure of diagnostic accuracy by sensitivity or specificity measurement. All other studies with incomplete information, insufficient material, or conflicting data were excluded from the review. In addition, it was done on the distribution by geography, number of studies by country, and frequency of studies per year.

### 2.5. Statistical Analysis

The extracted data were entered into a Microsoft Excel spreadsheet (version 19.0, Microsoft Corporation, Redmond, WA, USA) and then analyzed using the R programming environment (version 4.4.1) and its package “*mada*” (version 0.5.11) for meta-analysis of diagnostic accuracy (last access 23 July 2024). The “*mada*” package is used for meta-analysis of diagnostic accuracy studies. It estimates sensitivity, specificity, and likelihood ratios in a summary receiver operating characteristic curve for diagnostic tests. It also investigates the presence of heterogeneity across studies to arrive at appropriate conclusions regarding the diagnostic accuracy of medical tests [34,35].

The numbers of true negatives (TN), false negatives (FN), true positives (TP), and false positives (FP) were analyzed for each diagnostic test. Diagnostic accuracy was assessed by considering sensitivity and specificity. Sensitivity, or the true positive rate, is calculated as TP/(TP + FN), indicating the probability of a positive result in subjects with the disease. Specificity, or the true negative rate, is defined as TN/(TN + FP), representing the probability of a negative result in subjects without the disease.

The Positive Likelihood Ratio (LR+) measures the probability of a positive test result in diseased patients versus non-diseased, calculated as sensitivity/1-specificity. Values >10 indicate strong evidence of disease [36]. The Negative Likelihood Ratio (LR-) compares the probability of a negative result in diseased versus non-diseased patients, calculated as 1-sensitivity/specificity, with values <0.1 strongly indicating disease absence [37]. The Diagnostic Odds Ratio (DOR), combining LR+ and LR−, evaluates test effectiveness, where higher values indicate better diagnostic performance [36,38].

We used the model from “*Reitsma*” and its parameters from the “*mada*” package to obtain the summary receiver operating characteristic (sROC) curve, which estimates and compares the diagnostic performances of the tests [39]. This includes all sensitivity and specificity information obtained from individual studies to chart the sensitivity relationship with the false positive rate at different thresholds. Area under the curve (AUC) indicates how well a test performs overall, and greater AUC values reflect better diagnostic accuracy [40,41]. Also, the dispersion of study points around the sROC curve was judged visually for sources of heterogeneity. There was significant scattering in the case of high heterogeneity [42].

All calculations were carried out at a 95% confidence level to assure statistical validity, and the correction of continuity of 0.5 was used when required to make proper provision for small numbers of samples in the cells or cells with zero events to increase the accuracy of diagnostic performance metrics.

## 3. Results

### 3.1. Data Sources and Study Selection

This research conducted a systematic review and meta-analysis to evaluate the accuracy of diagnostic tests for rabies. A detailed flowchart outlining the study strategy was created and is displayed (Figure 1). To achieve this, a search using the MeSH terms “Rabies” and “Laboratory Diagnosis” was performed in the PubMed database, leading to the development of a MeSH term co-occurrence network map. The search identified 745 scientific studies published between 1990 and 2024.

The threshold for keyword occurrences was set to five, resulting in a network graph comprising 1352 MeSH keywords (Figure 2). The network map analysis reveals the formation of five primary clusters. The cluster associated with immunological diagnostic tests (green) includes terms such as “*Enzyme-Linked Immunosorbent Assay*” and “*Fluorescent Antibody Test*”. In the cluster about molecular diagnostic tests (purple, yellow), terms like “*Polymerase Chain Reaction*” and “*Reverse Transcriptase Polymerase Chain Reaction*” are prominent. Additionally, terms such as “*Rabies*”, “*Rabies virus*”, “*Antibodies*, *viral*”, “*Neutralization Tests*”, “*Humans*”, “*Dogs*”, and “*Brain*” were identified as common denominators (Figure 2). The terms identified during the initial analysis were employed in a secondary search within the PubMed database. These new search strings were formulated by integrating the newly identified terms with “*rabies*” and “*sensitivity and specificity*”, technical details see in Appendix A.

The number of retrieved studies on the performance of diagnostic tests for rabies was: 47 for Reverse Transcription Polymerase Chain Reaction (RT-PCR), 22 for Reverse Transcription Real-Time Polymerase Chain Reaction (RT-qPCR), 4 for Loop-mediated Isothermal Amplification (RT-LAMP), 1 for Clustered Regularly Interspaced Short Palindromic Repeats (CRISPR), 1 for Next Generation Sequencing (NGS), 21 for Rapid fluorescent focus inhibition test (RFFIT), 42 for Enzyme-linked Immunosorbent Assay (ELISA), 94 for Immunohistochemical Tests (IHT), 66 for Direct Fluorescent Antibody Test (DFAT), 53 for Immunochromatographic Assay (ICA), and 9 for Lateral Flow (LF). Our three-step selection criteria excluded 217 studies during the identification phase, 40 during the screening phase, and 43 during the eligibility phase. Consequently, 60 studies were included in the meta-analysis. Some of these studies reported multiple diagnostic tests, resulting in a total of 108 diagnostic studies included in the study (Figure 3). Regarding the geographical distribution of the studies, France, Brazil, and India had the highest number of studies related to diagnostic tests for rabies (Figure 3A). The number of studies by year is quite variable; recently, it was noted that the number of publications has a decreasing trend. Meanwhile, 2012 and 2020 had the highest number of publications (Figure 3B).

The methodological attributes of various laboratory tests for diagnosing rabies in humans and dogs were assessed. In humans, ELISA tests were predominantly utilized with antemortem serum samples across numerous studies, indicating their significant role in diagnostic applications. RT-PCR was identified as another widely used diagnostic method employed for both antemortem and postmortem samples, including brain tissue, saliva, and skin, thus offering comprehensive diagnostic coverage. DFAT was the principal reference test in multiple studies (Table 1).

Similarly, ELISA was commonly used in dogs for both antemortem and postmortem serum samples, with the Fluorescent Antibody Virus Neutralization Test (FAVNT) as the primary reference test. RT-PCR was also a key method, mainly applied postmortem to brain samples. The consistent use of DFAT and Mouse Inoculation Test (MIT) as reference standards underscore their essential role in confirming rabies diagnoses (Table 2).

### 3.2. Meta-Analysis of the Diagnostic Tests for Rabies

#### 3.2.1. Rabies in Humans

##### Enzyme-Linked Immunosorbent Assay

Eight studies were selected using the ELISA test [25,45,46,48,54,55,59,60]. A total of 2837 subjects were studied. Sensitivity ranged from 85.9 to 99.9%, with a median of 90.5%, 95%CI (77.0, 96.8); while the test for equality of sensitivities showed: χ^2^ = 57.94, df = 10, *p*-value = 8.86 × 10^−9^. Specificity ranged from 69.0 to 99.8%, with a median of 95.0%, 95%CI (84.9, 98.4); while the test for equality of specificities presented χ^2^ = 184.84, df = 10, *p*-value ≤ 2.00 × 10^−16^. The correlation between sensitivities and false positive rates was analyzed, and a negative result was shown: r = −0.485, 95%CI (−0.821, 0.223). In addition, results regarding LR+ {median 17.27, 95%CI (5.86, 61.10)}, LR− {median 0.10, 95%CI (0.03, 0.36)}, and DOR {median 201.00, 95%CI (20.30, 1476.64)} are displayed. The analyzed diagnostic performance is summarized in Figure 4 and Appendix A.

##### Reverse Transcription Polymerase Chain Reaction

Five studies based on the RT-PCR test were selected [43,44,47,57,58], in which 456 subjects were studied. Sensitivity ranged from 87.5% to 95.5%, with a median of 94.4%, 95%CI (62.9, 99.4), while the test for equality of sensitivities presented a χ^2^ = 0.41, df = 4, *p*-value = 0.982. Specificity ranged from 83.3 to 99.8%, with a median of 97.7%, 95%CI (81.6, 99.8); the test for equality of specificities showed χ^2^ = 27.69, df = 4, *p*-value = 1.44 × 10^−5^. A negative correlation between sensitivities and false positive rates is shown r = −0.765, 95%CI (–0.983, 0.361). Additionally, results regarding LR+ {median 41.56, 95%CI (3.61, 646.61)}, LR− {median 0.06, 95%CI (0.01, 0.84)}, and DOR {median 731.00, 95%CI (13.40, 39893.51)}. The analyzed diagnostic performance is summarized in Figure 5 and Appendix A.

##### Other Tests

For the diagnostic tests: Indirect Immunofluorescence Test (IIFT) [49], Latex Agglutination Test (LAT) [50], Dot Blot Enzyme Immunoassay (DBEI) [51], Rapid Neutralizing Antibody Test (RNAT) [56], Immunoperoxidase Inhibition Assay (IPIA) [27], Direct Rapid Immunohistochemical Test (DRIHT) [52], and FAVNT [45], only one study was included in the selection. Based on the established criteria, a minimum of five studies with a *p*-value of less than 0.05 were required for analysis. Consequently, no analysis was conducted for these diagnostic tests.

##### Summary ROC Curves (sROC)

A comparative analysis of data for human rabies diagnostic tests (ELISA and RT-PCR) was performed using an sROC curve (Figure 6). The observed differences in sensitivity and specificity are likely attributable to inherent or explicit variations between studies and differences in test cut-off points [98,99,100]. Figure 6 illustrates the area under the curve (AUC) for the rabies diagnostic tests, indicating the superior performance of ELISA. Additionally, both diagnostic tests demonstrated relatively high efficacy for detecting rabies in humans when the AUC was confined to the observed false positive rate (FPR) (AUC_FPR_) (Figure 6).

#### 3.2.2. Rabies in Dogs

##### Direct Fluorescent Antibody Test

Seven studies based on the DFAT test were selected [28,52,53,67,72,73,75], in which a total of 1226 subjects were studied. Sensitivity ranged from 40.9% to 99.7%, with a median of 79.2%, 95%CI (50.9, 93.3), while the test for equality of sensitivities presented a χ^2^ = 130.05, df = 28, *p*-value = 4.28 × 10^−15^. Specificity ranged from 25.0 to 99.7%, with a median of 95.0%, 95%CI (65.5, 99.5); the test for equality of specificities showed χ^2^ = 223.38, df = 28, *p*-value ≤ 2.00 × 10^−16^. A negative correlation between sensitivities and false positive rates is shown r = −0.056, 95%CI (–0.414, 0.317). Additionally, results regarding LR+ {median 13.64, 95%CI (0.91, 193.00)}, LR− {median 024, 95%CI (0.08, 0.81)}, and DOR {median 46.14, 95%CI (2.07, 1028.71)}. The analyzed diagnostic performance is summarized in Figure 7 and Appendix A.

##### Enzyme-Linked Immunosorbent Assay

Eight studies were selected using the ELISA test [69,75,80,93,94,95,97,101]. A total of 6654 subjects were studied. Sensitivity ranged from 54.2 to 98.0%, with a median of 88.9%, 95%CI (81.9, 92.4); while the test for equality of sensitivities showed: χ^2^ = 67.25, df = 9, *p*-value = 5.25 × 10^−11^. Specificity ranged from 95.0 to 99.6%, with a median of 99.2%, 95%CI (95.5, 99.7); while the test for equality of specificities presented χ^2^ = 40.14, df = 9, *p*-value = 7.15 × 10^−6^. The correlation between sensitivities and false positive rates was analyzed, and a negative result was shown: r = 0.225, 95%CI (−0.471, 0.749). In addition, results regarding LR+ {median 95.85, 95%CI (15.10, 344.45)}, LR− {median 0.11, 95%CI (0.08, 0.24)}, and DOR {median 463.39, 95%CI (174.87, 3742.70)} are displayed. The analyzed diagnostic performance is summarized in Figure 8 and Appendix A.

##### Rapid Immunochromatographic Tests

Twelve studies based on the RIT test were selected [61,71,76,77,78,81,86,87,88,89,92,96], in which a total of 3354 subjects were studied. Sensitivity ranged from 0.06% to 99.4%, with a median of 93.5%, 95%CI (83.7, 97.1), while the test for equality of sensitivities presented a χ^2^ = 718.06, df = 14, *p*-value ≤ 2.00 × 10^−16^. Specificity ranged from 91.6 to 99.7%, with a median of 99.1%, 95%CI (95.2, 99.9); the test for equality of specificities showed χ^2^ = 39.42, df = 14, *p*-value = 3.14 × 10^−4^. A negative correlation between sensitivities and false positive rates is shown r = 0.147, 95%CI (–0.395, −0.613). Additionally, results regarding LR+ {median 84.17, 95%CI (11.14, 1092.63)}, LR− {median 0.07, 95%CI (0.03, 0.18)}, and DOR {median 1235.61, 95%CI (82.26, 20,837.39)}. The analyzed diagnostic performance is summarized in Figure 9 and Appendix A.

##### Reverse Transcription Polymerase Chain Reaction

Eleven studies based on the RT-PCR test were selected [28,44,47,58,64,74,82,83,85,91,102], in which a total of 1356 subjects were studied. Sensitivity ranged from 66.4% to 99.5%, with a median of 94.4%, 95%CI (77.1, 98.7), while the test for equality of sensitivities presented a χ^2^ = 78.23, df = 14, *p*-value = 6.01 × 10^−11^. Specificity ranged from 83.3 to 99.5%, with a median of 98.6%, 95%CI (87.7, 99.9); the test for equality of specificities showed χ^2^ = 32.24, df = 14, *p*-value = 3.70 × 10^−3^. A negative correlation between sensitivities and false positive rates is shown r = −0.143, 95%CI (–0.611, 0.398). Additionally, results regarding LR+ {median 47.82, 95%CI (4.16, 753.87)}, LR− {median 0.06, 95%CI (0.01, 0.31)}, and DOR {median 309.56, 95%CI (21.32, 5395.31)}. The analyzed diagnostic performance is summarized in Figure 10 and Appendix A.

##### Other Tests

Regarding the diagnostic tests: Immunohistochemical Tests (IHT), Rapid Immunodiagnostic Assay (RIA), and Immunoperoxidase Tests (IPT), three studies [52,65,66], three studies [79,90,91], and two studies [62,84] were selected, respectively. Additionally, for the diagnostic tests DBEI [51], FAVNT [68], RFFIT [63], RT-qPCR [28], RT-LAMP [83], and RT-RPA [70], only one study was included in the selection. Analysis was recommended to be carried out on these diagnostic tests, based on the set criteria of at least five qualifying studies whose *p*-values were less than 0.05. As a result, analysis could not be done for these diagnostic techniques because no study qualified for inclusion.

##### Summary ROC Curves (sROC)

Rabies diagnostic tests in dogs (DFAT, ELISA, RIT, and RT-PCR) were evaluated using a summary receiver operating characteristic (sROC) curve analysis (Figure 11). Variations in sensitivity and specificity were attributed to implicit and explicit differences among the studies and variations in test cut-off points [98,99,100]. Figure 11 illustrates the calculated area under the curve (AUC) for these rabies diagnostic tests, highlighting the superior performance of RIT and ELISA. Additionally, when the AUC was constrained to the observed false positive rate (FPR), the RIT diagnostic test exhibited satisfactory performance for rabies detection (AUC_FPR_) (Figure 11).

## 4. Discussion

### 4.1. Summary of Main Findings

There are significant concentrations of studies in France, Brazil, and India. This suggests that as well as providing resources for research, these areas are crucial for advancing diagnosis research because of the high prevalence of rabies. The high prevalence of rabies in Africa and Asia, along with the increasing prevalence in South America in several epidemiological studies, enforces continuous research and upgrades the diagnostic tools of countries like India and Brazil [4]. Furthermore, research institutions and funding organizations in such countries are established. For instance, the Institute Pasteur in France has been a powerhouse for many years, in terms of research on rabies and discoveries that give critical insights into diagnosis [103]. The interest in rabies research and its resulting output, including the number of studies and patents, varies markedly across different regions. Regions severely impacted by rabies frequently lack the infrastructure to undertake independent research, underscoring the critical need for international collaboration and support. Such international cooperation is pivotal in enhancing local research capacities and bolstering diagnostic capabilities within these affected areas [104,105,106]. Collaborative efforts between well-established institutions and those in resource-limited settings can significantly facilitate knowledge transfer and technology, thereby improving local rabies response [107]. Focusing rabies research on specific regions highlights the existing disparities in research capabilities and emphasizes the global responsibility to address these gaps. Through increased international cooperation and support, advancements in rabies diagnosis and treatment can be made accessible to all regions, particularly those most severely affected by the disease [108,109].

Furthermore, the temporal analysis shows a variation in the number of publications throughout the years: one can observe a disturbingly decreasing tendency in the last. This might be related to a change in research orientation, lack of economic resources, or the assumption of improvement of diagnostic technologies, among other issues that might increase the interest in this matter [110,111]. However, only two peaks were of interest in 2012 and 2020, which could be due to special events, including outbreaks, improvements in diagnostic technology, or targeted research initiatives. This 2012 peak could very well be related to the increased attention after the 2010 WHO report, which pointed out the global burden of rabies [112]. The most likely explanation for the increase in 2020 is that people were confined, leading to a year in which more studies were written than research was conducted. This period allowed many to take advantage of the challenges posed by remote work to complete and publish manuscripts that had been previously set aside [113].

Our observations indicate that the majority of diagnostic tests are conducted post-mortem. Consequently, there is a critical need to enhance the frequency of antemortem diagnostic tests to facilitate early detection and effective disease management. Such advancements have the potential to improve patient outcomes markedly and decrease mortality rates [114]. Despite significant progress in medical technology, current diagnostic practices often depend predominantly on post-mortem confirmations, thereby restricting opportunities for timely medical interventions [28]. By increasing the availability and accuracy of antemortem tests, we can improve clinical decision-making and gain a more comprehensive understanding of disease progression and epidemiology [115].

### 4.2. Rabies in Humans

Data robusticity is guaranteed because 2837 subjects were enrolled in eight studies that analyzed the use of ELISA for diagnosing rabies in humans, all giving promising results. The range of sensitivity between 85.9% and 99.9%, with a median of 90.5%, proves the high efficacy of ELISA at properly identifying patients with rabies. Similarly, the specificity range of 69.0–99.8%, with a median value of 95.0%, suggests that the test is also suitable for correctly detecting those who do not have the disease. These high sensitivity and specificity values are crucial for any diagnostic test, since they guarantee the test’s dependability in differentiating between infected and noninfected people. [116]. The large heterogeneity found in sensitivity implies variability across studies, which may be due to differences in study design, population, or test implementation [117]. In most instances, the consistent general performance obtained from a large number of subjects offers reassurance. There is a negative correlation between sensitivities and false positive rates, i.e., as the ability of the test to detect true positives increases, the rate of false positives decreases, further supporting its reliability [118]. These are also adequate diagnostic metrics, with an LR+ of 17.27, LR− of 0.10, and a DOR of 201.00. A high LR+ shows that it is very likely for a positive test result in a person with rabies compared to one without, whereas a low LR− shows a negative test result much less likely in a person with rabies [119]. The DOR takes all these ratios and reflects the high accuracy of ELISA. On the other hand, the five studies with 456 subjects using RT-PCR for diagnosis of rabies had a high degree of diagnostic accuracy with sensitivity ranging between 87.5% and 95.5% and specificity ranging between 83.3% and 99.8%, with median values of 94.4% and 97.7%. While there was a significant heterogeneity in specificity, sensitivity did not show any inconsistency. This indicates the reliability of the test, because a strong negative correlation of sensitivity with false positive rates is shown. High LR+ and low LR−, associated with a DOR of 731.00, show high efficacy of RT-PCR in rabies detection.

The comparative assessment of ELISA and RT-PCR for diagnosing human rabies, utilizing sROC curves, identifies variations in sensitivity and specificity due to inherent study differences and varying test cut-off points. ELISA’s superior performance, demonstrated by a higher AUC, may be associated with its consistent diagnostic reliability across diverse conditions. Factors such as study design, population characteristics, methodologies, and specific testing thresholds significantly impact diagnostic accuracy [120,121]. Both diagnostic methods showed high efficiency in detecting rabies when the AUC_FPR_ was used, underscoring their overall reliability despite observed differences [122]. The high efficiency of ELISA is due to its consistent performance in detecting antibodies against the rabies virus across various conditions, leading to high sensitivity and specificity, as demonstrated by a superior AUC in sROC analyses [93].

Similarly, RT-PCR’s efficiency stems from its direct detection of viral RNA, which maintains high sensitivity, even with low viral loads. This efficiency is further validated by high positive likelihood ratios (LR+) and low negative likelihood ratios (LR−), showcasing its strong ability to confirm positive cases and rule out negative ones [123]. Collectively, these diagnostic metrics highlight the overall high reliability and accuracy of both tests in clinical settings.

The scarcity of studies on human rabies diagnostic techniques, including IIFT, LAT, DBEI, RNAT, IPIA, DRIHT, and FAVNT, can be attributed to several factors. The global burden of rabies predominantly affects marginalized populations in regions where resources for extensive human studies are limited [124]. Furthermore, diagnostic efforts often prioritize animal models due to the higher prevalence and easier study conditions of rabies in animal populations [108]. Financial and logistical constraints and ethical considerations further restrict the scope and number of human studies in this area [125].

### 4.3. Rabies in Dogs

In an analysis spanning seven studies and encompassing 1226 subjects, the DFAT exhibited a broad range of sensitivity from 40.9% to 99.7% and specificity from 25.0% to 99.7%, with significant heterogeneity noted in both parameters. The negative correlation between sensitivity and false positive rates (r = −0.056) alongside diagnostic metrics such as LR+ (13.64), LR− (0.24), and DOR (46.14) highlight the variability and diagnostic challenges associated with DFAT. In contrast, ELISA demonstrated greater consistency in its diagnostic performance across eight studies involving 6654 subjects, with sensitivity ranging from 54.2% to 98.0% and specificity from 95.0% to 99.6%, supported by robust diagnostic metrics: LR+ (95.85), LR− (0.11), and DOR (463.39). RIT, evaluated in twelve studies with 3354 subjects, also showed high diagnostic accuracy, with sensitivity ranging from 0.06% to 99.4% and specificity from 91.6% to 99.7%, complemented by solid metrics: LR+ (84.17), LR− (0.07), and DOR (1235.61). Finally, RT-PCR, assessed across eleven studies involving 1356 subjects, demonstrated sensitivity from 66.4% to 99.5% and specificity from 83.3% to 99.5%, with solid diagnostic metrics: LR+ (47.82), LR− (0.06), and DOR (309.56), underscoring its high diagnostic accuracy and reliability for rabies detection. RIT, RT-PCR, and ELISA are more consistent and reliable than DFAT due to their higher and more stable diagnostic accuracy, demonstrated by strong and consistent sensitivity and specificity across multiple studies. These methods also exhibit robust diagnostic metrics such as high LR+, low LR−, and high DOR, indicating a high probability of correctly identifying both infected and non-infected subjects. RIT, RT-PCR, and ELISA also show minimal variability and heterogeneity in their diagnostic parameters, ensuring dependable performance across different study designs and populations. In contrast, DFAT exhibits significant variability and heterogeneity, leading to less consistent and reliable diagnostic outcomes. The assessment of rabies diagnostic tests in canines, encompassing DFAT, ELISA, RIT, and RT-PCR, conducted through sROC curve analysis, indicated notable disparities in sensitivity and specificity attributed to study differences and test cut-off points. The analysis demonstrated the superior performance of RIT and ELISA, as reflected by the AUC. Notably, RIT exhibited satisfactory performance when evaluated through AUC_FPR_. Supporting this data is another report indicating that traditional techniques, such as FAT, may not always be optimal and produce false negative results under certain conditions, such as low viral load [24].

Traditionally regarded as the gold standard for rabies diagnosis, DFAT presents several limitations impacting its reliability and practicality. Notable concerns include variability in test results due to inconsistent antigen localization within brain tissues and the quality of the immunofluorescent conjugate. Such variability can lead to false negatives, particularly in low viral load samples or when procedural standards are not rigorously adhered to [28,79]. DFAT’s requirement for sophisticated equipment and skilled personnel also restricts its use in resource-limited settings. In many developing countries, deviations from standard protocols, such as the use of expired reagents and lack of quality controls, further undermine its accuracy [126]. Additionally, a major limitation of studies employing DFAT is that they are predominantly conducted on post-mortem samples, which suggests that the diagnostic approaches optimized for post-mortem conditions may not be directly applicable or as effective in antemortem scenarios [115]. Conversely, alternative methods like RIT, RT-PCR, and ELISA have demonstrated more consistent and reliable performance with minimal variability and heterogeneity across diverse studies and populations [115,127].

### 4.4. Strengths and Limitations

The strengths of this scientific article are evident through its rigorous methodology and comprehensive analysis. The use of a systematic search strategy employing the MeSH terms “*Rabies*” and “*Laboratory Diagnosis*”, along with terms related to diagnostic tests, in the PubMed database resulted in the identification of 360 studies published between 1990 and 2024, establishing a robust dataset for analysis [128]. The meticulous three-step selection process ensured the inclusion of only relevant and high-quality studies, enhancing the reliability of the meta-analysis findings [129]. By encompassing a diverse range of diagnostic tests for rabies in humans and dogs, such as ELISA, RT-PCR, DFAT, and RIT, the study thoroughly evaluates various diagnostic methods and their respective accuracies [130]. The application of meta-analytic techniques and sROC curves further strengthens the study by facilitating a comparative analysis of the diagnostic performance of various tests [131,132]. Comprehensive metrics, including sensitivity, specificity, LR+, LR−, and DOR, are thoroughly analyzed, presenting a clear and detailed picture of test efficacy, thus contributing valuable insights to the field of rabies diagnosis [132].

The study’s reliance exclusively on the PubMed database, while comprehensive in scope, may have restricted its dataset by not incorporating other major databases such as Embase, Scopus, and Web of Science. This exclusion could have led to the omission of relevant studies, thereby narrowing the breadth of the dataset and potentially overlooking important research that could have enriched the analysis [133]. Additionally, the evaluation of specific diagnostic tests—such as IIFT, LAT, DBEI, RNAT, IPIA, DRIHT, and FAVNT—was limited because only one study per test was included. This restriction hindered a thorough assessment due to the requirement for at least five studies to achieve a robust and statistically reliable analysis [134]. Furthermore, the variability in the quality of the included studies, as evidenced by discrepancies in sensitivity and specificity, may compromise the reliability of the meta-analysis findings.

Factors such as differences in study design, sample sizes, and test cut-off points contributed to this variability, potentially affecting the consistency and validity of the results [135]. The observed decline in the number of publications in recent years, contrasted with peaks in 2012 and 2020, raises concerns about potential publication bias. This trend suggests that studies with significant or positive findings are more likely to be published, which could skew the meta-analysis outcomes and affect the generalizability of the conclusions [136]. Moreover, although the observed negative correlation between sensitivities and false positive rates provides initial insights, the analysis did not fully explore the underlying reasons for this relationship. This analysis also did not thoroughly investigate potential factors that might contribute to the variability in results, such as the inherent trade-offs between sensitivity and specificity in diagnostic tests, where an increase in sensitivity often leads to higher false positive rates and vice versa. Additionally, variations in cut-off values or thresholds, which can impact both sensitivity and false positive rates, and the effect of disease prevalence on test accuracy, were also not examined. Differences in assay techniques, sample quality, and procedural variations could further contribute to discrepancies in test results. The limitations related to the substantial variability in sample types and conditions (antemortem and postmortem) used in the studies must also be considered, as this heterogeneity presents significant challenges in establishing a consistent relationship between these factors and the diagnostic accuracy of the evaluated techniques. The inclusion of various sample types, such as cerebrospinal fluid, saliva, and brain tissue, each with distinct characteristics and varying levels of degradation, can lead to inconsistencies in diagnostic outcomes. For instance, postmortem samples may exhibit different degrees of decomposition, adversely impacting the performance of diagnostic assays, such as RT-PCR and DFAT, both of which are highly sensitive to sample integrity. Furthermore, the influence of study design factors, such as sample size, demographic characteristics, and the statistical approaches used in data analysis, might provide a deeper understanding of this correlation. A more detailed exploration of these factors could have yielded a better understanding of the diagnostic tests’ reliability and a more precise assessment of their performance [137,138,139].

### 4.5. Implications for Future Research

Given the rapid progress in diagnostic tests, future research into rabies diagnosis should incorporate state-of-the-art methods to tackle existing challenges and boost diagnostic accuracy. NGS provides a detailed approach for detecting the genetic material of the rabies virus, allowing for precise identification of viral variants and mutations, surpassing the capabilities of traditional methods [140]. This technology offers an in-depth genomic analysis that could be crucial for understanding the virus’s evolution and epidemiology. Furthermore, high-throughput immunoassays, including multiplex assays, enable the simultaneous assessment of various biomarkers and antibodies, delivering fast and reliable results that improve diagnostic efficiency and shorten processing times [141]. Additionally, these innovative diagnostic techniques have significantly expanded the capabilities of rabies diagnosis by analyzing a diverse array of sample types, including non-traditional ones such as saliva and urine, which are minimally invasive [24]. The ability to use these non-traditional samples facilitates more accessible and more frequent testing, particularly in resource-limited settings where traditional sample collection methods may be challenging. As a result, implementing these advanced techniques holds great promise for improving rabies surveillance, early detection, and timely intervention, ultimately contributing to better control and prevention of this deadly disease [142,143].

The analysis of diagnostic tests for rabies reveals significant variability in their performance, pointing to a crucial need for validating new diagnostic tools in a range of different settings [22]. This means that current tests may not work equally well in all environments or situations, leading to inconsistent accuracy in detecting the disease [144]. Introducing advanced diagnostic technologies could help overcome these limitations. For example, newer methods could offer more accurate and faster results, which would enhance the reliability of diagnoses [143]. This improvement is essential for timely and effective treatment, ultimately leading to better health outcomes for humans and dogs by ensuring the disease is detected and managed more efficiently.

## 5. Conclusions

The diagnosis of rabies in humans and dogs poses significant challenges due to the inconsistent performance of current diagnostic methods. This systematic review and me-ta-analysis underscore the efficacy of immunological tests (ELISA) and molecular tests (RT-PCR) in humans, as well as immunological (RIT) and molecular (RT-PCR) tests in dogs. Variations in sensitivity and specificity are attributed to differences in study methodologies and test cut-off points. Although the DFAT has long been regarded as the gold standard for directly detecting the rabies virus in brain tissue, its diagnostic accuracy is constrained, potentially due to variability in antigen distribution within brain tissues and the quality of the immunofluorescent conjugate. Such limitations may lead to false negatives, particularly in samples with low viral loads or when procedural rigor is lacking. These issues underscore the necessity to reevaluate and update rabies diagnostic protocols by incorporating advanced technological approaches. Integrating novel diagnostic techniques that offer enhanced speed, precision, and user-friendliness could markedly improve outbreak management and decrease rabies mortality, particularly in endemic regions, enabling more timely and effective interventions and better control of viral transmission.

## Figures and Tables

**Figure 1 diagnostics-15-00412-f001:**
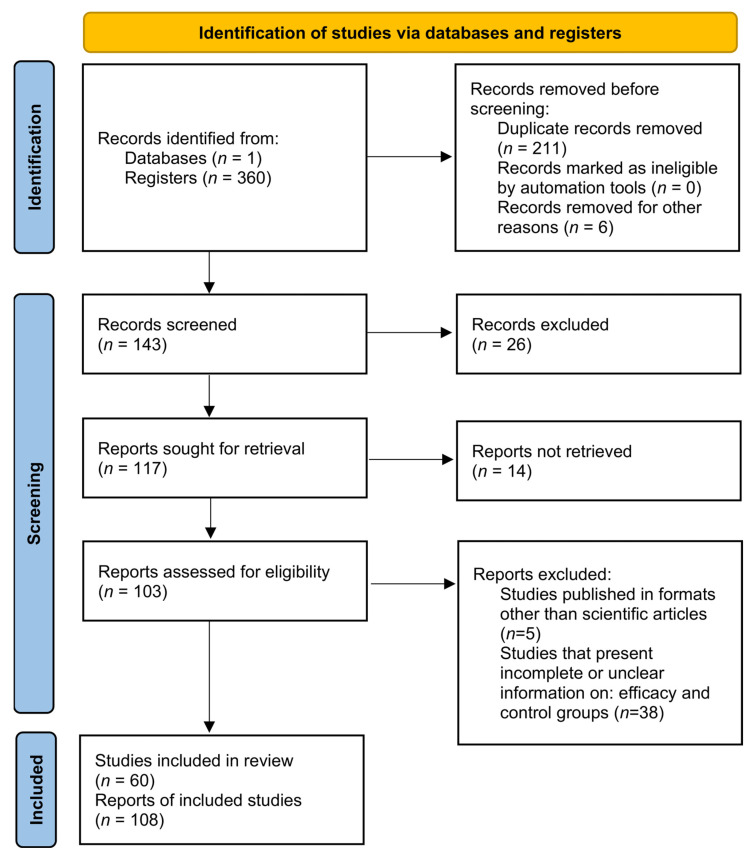
A systematic review and meta-analysis flowchart detailing the study selection process.

**Figure 2 diagnostics-15-00412-f002:**
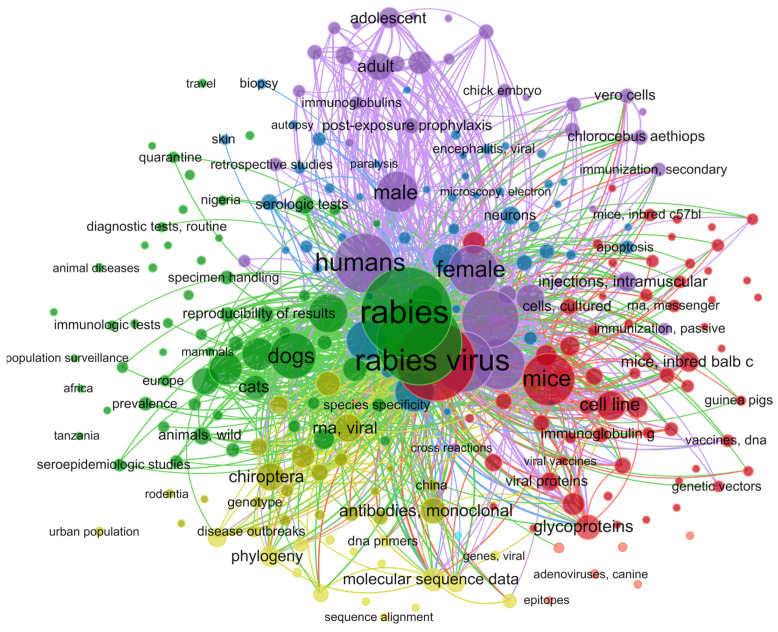
A bibliometric map was generated using VOSviewer, illustrating the co-occurrence of MeSH terms in the studies selected for various rabies diagnostic techniques.

**Figure 3 diagnostics-15-00412-f003:**
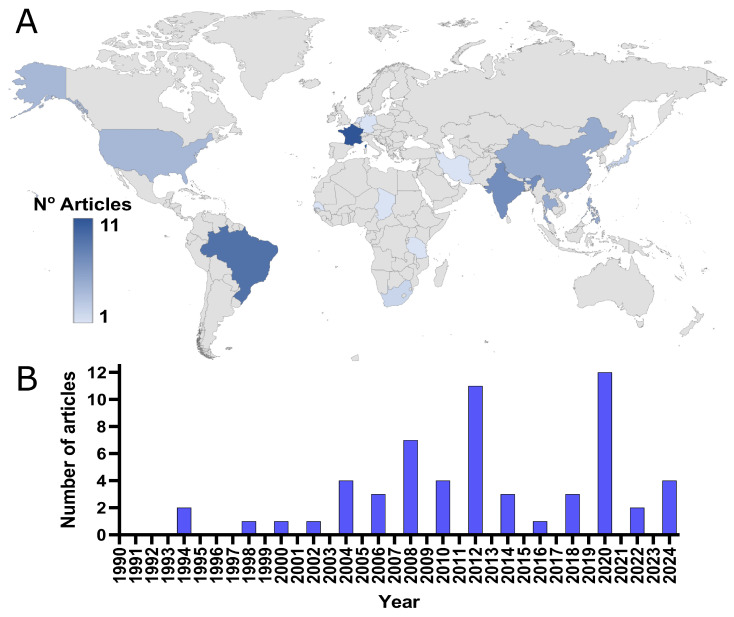
Mapping and temporal trends of rabies diagnostic research. (**A**) Geographical distribution: The map represents the number of rabies diagnostic studies published per country. Countries are shaded according to the number of studies, ranging from 1 (light blue) to 11 (dark blue). (**B**) Temporal trend: The bar chart shows the annual number of rabies-diagnostic studies published between 1990 and 2024.

**Figure 4 diagnostics-15-00412-f004:**
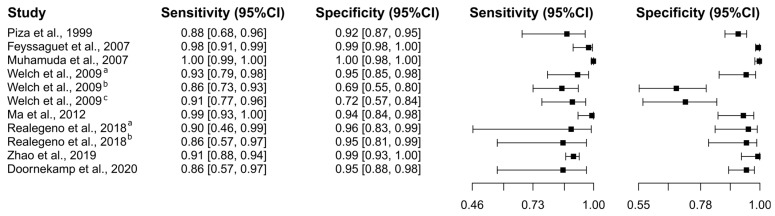
Study data and paired forest plot of the sensitivity and specificity of Enzyme-Linked Immunosorbent Assay (ELISA) for human rabies diagnosis. Data from each study are summarized. Sensitivity and specificity are reported with a mean (95% confidence limits). The Forest plot depicts the estimated sensitivity and specificity (black squares) and its 95% confidence limits (horizontal black line) [25,45,46,48,54,55,59,60].

**Figure 5 diagnostics-15-00412-f005:**
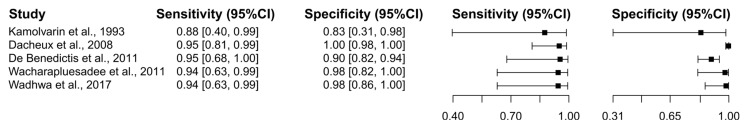
Study data and paired forest plot of the sensitivity and specificity of Reverse Transcription Polymerase Chain Reaction (RT-PCR) for human rabies diagnosis. Data from each study are summarized. Sensitivity and specificity are reported with a mean (95% confidence limits). The Forest plot depicts the estimated sensitivity and specificity (black squares) and its 95% confidence limits (horizontal black line) [43,44,47,57,58].

**Figure 6 diagnostics-15-00412-f006:**
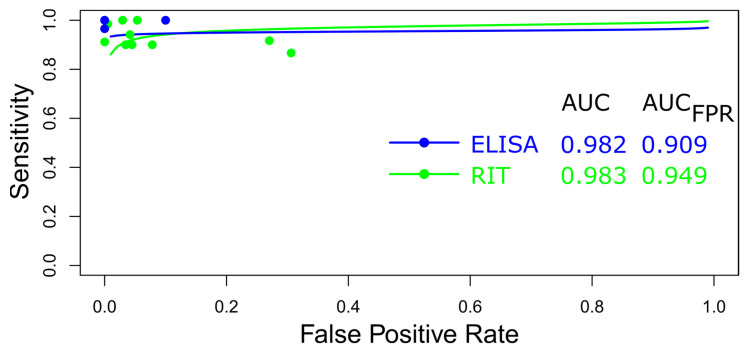
Meta-analysis of diagnostic test accuracy analysis. Summary receiver operating curve (sROC) plot of false positive rate and sensitivity. Comparison between ELISA and RT-PCR methods in the diagnosis of rabies in humans.

**Figure 7 diagnostics-15-00412-f007:**
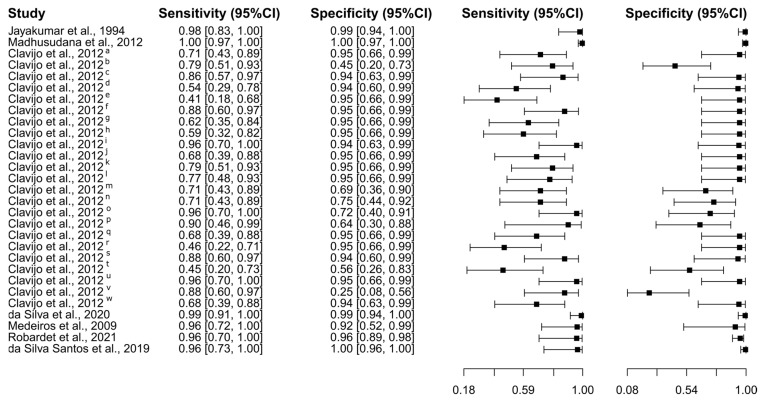
Study data and paired forest plot of the sensitivity and specificity of Direct Fluorescent Antibody Test (DFAT) for rabies diagnosis in dogs. Data from each study are summarized. Sensitivity and specificity are reported with a mean (95% confidence limits). The Forest plot depicts the estimated sensitivity and specificity (black squares) and its 95% confidence limits (horizontal black line) [28,52,53,67,72,73,75].

**Figure 8 diagnostics-15-00412-f008:**
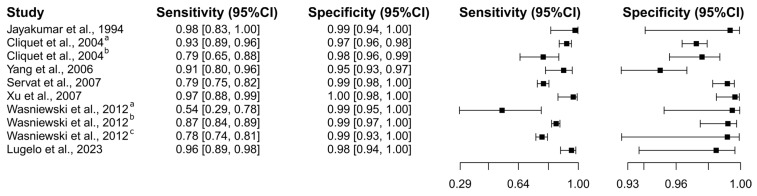
Study data and paired forest plot of the sensitivity and specificity of Enzyme-Linked Immunosorbent Assay (ELISA) for rabies diagnosis in dogs. Data from each study are summarized. Sensitivity and specificity are reported with a mean (95% confidence limits). The Forest plot depicts the estimated sensitivity and specificity (black squares) and its 95% confidence limits (horizontal black line) [69,75,80,93,94,95,97,101].

**Figure 9 diagnostics-15-00412-f009:**
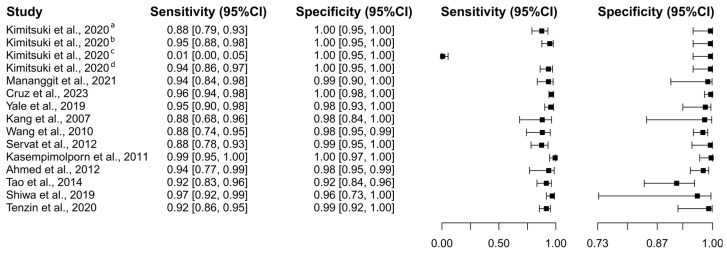
Study data and paired forest plot of the sensitivity and specificity of Rapid Immunochromatographic Tests (RIT) for dog rabies diagnosis. Data from each study are summarized. Sensitivity and specificity are reported with a mean (95% confidence limits). The Forest plot depicts the estimated sensitivity and specificity (black squares) and its 95% confidence limits (horizontal black line) [61,71,76,77,78,81,86,87,88,89,92,96].

**Figure 10 diagnostics-15-00412-f010:**
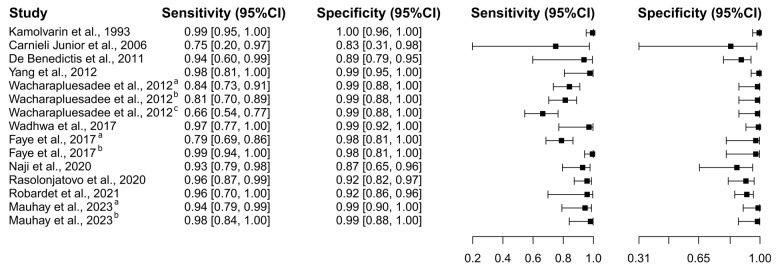
Study data and paired forest plot of the sensitivity and specificity of Reverse Transcription Polymerase Chain Reaction (RT-PCR) for rabies diagnosis in dogs. Data from each study are summarized. Sensitivity and specificity are reported with a mean (95% confidence limits). The Forest plot depicts the estimated sensitivity and specificity (black squares) and its 95% confidence limits (horizontal black line) [28,44,47,58,64,74,82,83,85,91,102].

**Figure 11 diagnostics-15-00412-f011:**
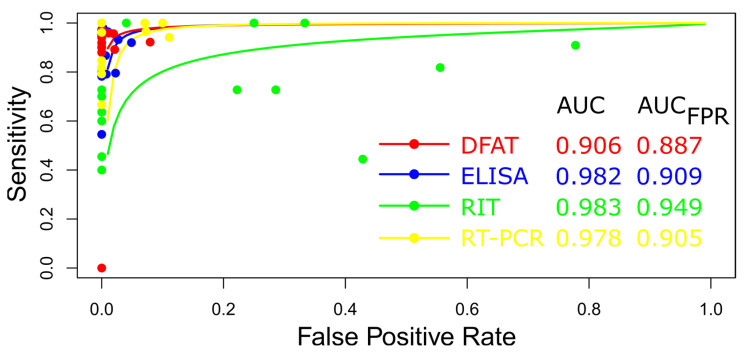
Meta-analysis of diagnostic test accuracy analysis. Summary receiver operating curve (sROC) plot of false positive rate and sensitivity. Comparison between DFAT, ELISA, RIT, and RT-PCR methods in diagnosing rabies in dogs.

**Table 1 diagnostics-15-00412-t001:** Main methodological characteristics of studies addressing the diagnosis of rabies in humans.

Reference	Diagnostic Test	Sample Size	Type of Sample	Mortality of Sample	Reference Test
Batista et al., 2011 [27]	IPIA	422	Serum	Antemortem	MIT
Dacheux et al., 2008 [43]	RT-PCR	285	Skin, saliva, urine, serum and brain	Antemortem and Postmortem	DFAT
De Benedictis et al., 2011 [44]	RT-PCR	100	Skin, saliva and brain	Antemortem and Postmortem	DFAT
Doornekamp et al., 2020 [45]	ELISA, FAVNT	99	Serum and dried blood spots	Postmortem	FAVNT
Feyssaguet et al., 2007 [46]	ELISA	655	Serum	Antemortem	RFFIT
Kamolvarin et al., 1993 [47]	RT-PCR	5	Brain	Postmortem	DFAT and MIT
Ma et al., 2012 [48]	ELISA	120	Serum and cerebrospinal fluid	Antemortem	RFFIT
Madhusudana et al., 2001 [49]	IIFT	193	Serum and cerebrospinal fluid	Antemortem	MIT
Madhusudana et al., 2003 [50]	LAT	229	Serum	Antemortem	MIT
Madhusudana et al., 2004 [51]	DBEI	115	Brain, Cerebrospinal fluid and saliva	Antemortem and Postmortem	DFAT and MIT
Madhusudana et al., 2012 [52]	DFAT, DRIHT	38	Brain	Postmortem	DFAT
Medeiros et al., 2009 [53]	DFAT	8	Central nervous system	Postmortem	DFAT
Muhamuda et al., 2007 [54]	ELISA	990	Serum	Antemortem	RFFIT
Piza et al., 1999 [55]	ELISA	199	Serum	Antemortem	FAVNT
Realegeno et al., 2018 [25]	ELISA	38	Serum and cerebrospinal fluid	Antemortem	IIFT
Shiota et al., 2009 [56]	RNAT	115	Serum	Antemortem	RFFIT
Wacharapluesadee et al., 2011 [57]	RT-PCR	29	Saliva, cerebrospinal fluid, urine, hair and brain	Antemortem and Postmortem	DFAT and MIT
Wadhwa et al., 2017 [58]	RT-PCR	37	Brain	Antemortem and Postmortem	DFAT
Welch et al., 2009 [59]	ELISA	94	Serum	Antemortem	RFFIT
Zhao et al., 2019 [60]	ELISA	415	Serum	Antemortem	RFFIT

**Table 2 diagnostics-15-00412-t002:** Main methodological characteristics of studies addressing the diagnosis of rabies in dogs.

Reference	Diagnostic Test	Sample Size	Type of Sample	Mortality of Sample	Reference Test
Ahmed et al., 2012 [61]	RIT	228	Brain	Postmortem	DFAT
Arslan et al., 2004 [62]	IPT	81	Brain	Postmortem	DFAT
Cardoso et al., 2006 [63]	RFFIT	211	Serum	Antemortem	MIT
Carnieli et al., 2006 [64]	RT-PCR	3	Brain	Postmortem	DFAT and MIT
Castro et al., 2020 [65]	IHT	32	Brain	Postmortem	DFAT
Claassen et al., 2023 [66]	IHT	199	Brain	Postmortem	DFAT
Clavijo et al., 2017 [67]	DFAT	20	Brain	Postmortem	DFAT and RTCIT
Cliquet et al., 1998 [68]	FAVNT	414	Serum	Antemortem	FAVNT and RFFIT
Cliquet et al., 2004 [69]	ELISA	2360	Serum	Antemortem	FAVNT
Coertse et al., 2019 [70]	RT-RPA	109	Brain	Postmortem	DFAT and IHT
Cruz et al., 2023 [71]	RIT	791	Brain	Postmortem	DFAT
da Silva et al., 2020 [72]	DFAT	125	Central nervous system	Postmortem	DFAT
da Silva Santos et al., 2019 [73]	DFAT	125	Brain	Postmortem	DFAT
De Benedictis et al., 2011 [44]	RT-PCR	67	Brain and saliva	Postmortem	DFAT
Faye et al., 2017 [74]	RT-PCR	97	Brain	Postmortem	DFAT
Jayakumar et al., 1994 [75]	DFAT, ELISA	100	Brain	Postmortem	DFAT
Kamolvarin et al., 1993 [47]	RT-PCR	205	Brain	Postmortem	DFAT and MIT
Kang et al., 2007 [76]	RIT	51	Brain and saliva	Postmortem	IFAT
Kasempimolporn et al., 2011 [77]	RIT	237	Saliva	Antemortem	DFAT
Kimitsuki et al., 2020 [78]	RIT	184	Brain	Postmortem	DFAT
Léchenne et al., 2016 [79]	RIA	45	Brain	Postmortem	DFAT
Lugelo et al., 2023 [80]	ELISA	201	Serum	Antemortem	FAVNT
Madhusudana et al., 2004 [70]	DBEI	210	Brain, Cerebrospinal fluid and saliva	Antemortem and Postmortem	DFAT and MIT
Madhusudana et al., 2012 [52]	DFAT, IHT	320	Brain	Postmortem	DFAT
Mananggit et al., 2021 [81]	RIT	97	Brain	Postmortem	DFAT
Mauhay et al., 2023 [82]	RT-PCR	130	Brain	Postmortem	DFAT
Medeiros et al., 2009 [53]	DFAT	17	Central nervous system	Postmortem	DFAT
Naji et al., 2019 [83]	RT-PCR, RT-LAMP	50	Brain	Postmortem	DFAT
Ogawa et al., 2008 [84]	IPT	310	Serum	Antemortem	FAVNT
Rasolonjatovo et al., 2020 [85]	RT-PCR	113	Brain	Postmortem	DFAT
Robardet et al., 2021 [28]	DFAT, RT-PCR, RT-qPCR	110	Brain	Postmortem	DFAT and MIT
Servat et al., 2012 [86]	ELISA, RIT	172	Brain	Postmortem	DFAT and RTCIT
Shiwa et al., 2019 [87]	RIT	123	Brain and skin	Postmortem	DFAT
Tao et al., 2014 [88]	RIT	165	Serum	Antemortem	ELISA
Tenzin et al., 2020 [89]	RIT	179	Brain	Antemortem and Postmortem	DFAT
Voehl et al., 2014 [90]	RIA	104	Brain	Postmortem	DFAT
Wacharapluesadee et al., 2012 [91]	RT-PCR	101	Oral swab and hair	Postmortem	DFAT
Wadhwa et al., 2017 [58]	RT-PCR	71	Brain, skin, saliva and cornea	Postmortem	DFAT
Wang et al., 2010 [92]	RIT	366	Serum	Postmortem	FAVNT
Wasniewski et al., 2012 [93]	ELISA	1123	Serum	Antemortem	FAVNT and RFFIT
Wasniewski et al., 2014 [94]	ELISA	593	Serum	Antemortem	FAVNT
Xu et al., 2007 [95]	ELISA	475	Brain	Postmortem	DFAT and RTCIT
Yale et al., 2019 [96]	RIT	209	Brain	Postmortem	DFAT
Yang et al., 2006 [97]	ELISA	500	Serum	Antemortem	FAVNT
Yang et al., 2012 [91]	RIA, RT-PCR	110	Brains	Postmortem	DFAT

## Data Availability

Not applicable.

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
