# Peer review of "Evaluating Rabies Test Accuracy: A Systematic Review and Meta-Analysis of Human and Canine Diagnostic Methods"

_diagnostics, 2025, doi:10.3390/diagnostics15040412_

Round 1
Reviewer 1 Report
Comments and Suggestions for Authors
This systematic review with meta-analysis provides valuable insights into the diagnosis of rabies. I commend the authors for their solid methodological approach and execution. It also clearly indicates the limiting factors of the study. The presented work will contribute to the study of the virus, and I strongly recommend its publication. I think three minor comments/suggestions will improve the manuscript.
1. In Materials and Methods, I find a bit excessive the description of terms like sensitivity, specificity, etc.
2. Line 226: Remove "immunological", since PCR, etc. are not immunological tests.
3. I suggest revising Tables 1 and 2 in chronological, alphabetic, or some other intuitive order.
Author Response
This systematic review with meta-analysis provides valuable insights into the diagnosis of rabies. I commend the authors for their solid methodological approach and execution. It also clearly indicates the limiting factors of the study. The presented work will contribute to the study of the virus, and I strongly recommend its publication. I think three minor comments/suggestions will improve the manuscript.
Answer: We thank the reviewer for your kind and supportive comments.
- In Materials and Methods, I find a bit excessive the description of terms like sensitivity, specificity, etc.
Answer: Thank you for your observation; the description of these terms has been revised and condensed.
- Line 226: Remove "immunological", since PCR, etc. are not immunological tests.
Answer: Thank you for your observation; the term has been removed.
- I suggest revising Tables 1 and 2 in chronological, alphabetic, or some other intuitive order.
Answer: Thank you for your valuable observation. The suggested change has been implemented, and the tables have been reorganized in alphabetical order.
Reviewer 2 Report
Comments and Suggestions for Authors
1. It is important that the abbreviations of the tests used in the article are explained in their first place so that the reader can easily understand them. 2. Providing information about leishmania in the results section (225-229) is not considered appropriate for the integrity of the text. This information can be presented appropriately in the discussion section.
Author Response
- It is important that the abbreviations of the tests used in the article are explained in their first place so that the reader can easily understand them.
Answer: Thank you for your observation. The suggested changes regarding the abbreviations have been implemented, and all test abbreviations are now explained at their first mention to enhance the reader's understanding.
- Providing information about leishmania in the results section (225-229) is not considered appropriate for the integrity of the text. This information can be presented appropriately in the discussion section.
Answer: Thank you for your observation. There was an error in the text, and the data referenced do not pertain to leishmaniasis but rather to rabies. The correction has been made accordingly.
Reviewer 3 Report
Comments and Suggestions for Authors
This review article targets an important subject: rabies diagnostics. This topic is a great match to this journal, and interesting for the readers. The authors are encouraged to address the comments below.
The authors are encouraged to polish the entire manuscript to maintain the accuracy. For example, in the manuscript title, the comprehensive and systemic are redundant, and you need only one of these two words; the writing of "human and canine diagnostics" is very confusing. You can consider modifying the title as: "Evaluating Rabies Test Accuracy: A Systematic Review and Meta-Analysis of Human and Canine Diagnostic Methods."
Also in the manuscript abstract, the text reads "Compared to the direct fluorescent antibody test (DFAT), the area under the curve for false positive rates (AUCFPR= 0.887) 32 showed lower accuracy." I do not understand this sentence, and you are comparing the accuracy between which assays.
All the figures need to have a figure legend. For example, for a busy figure 3, the readers can understand this busy figure without a figure legend.
As a review article, please include the newest references. I see that there is only one 2024 reference here. Please add more recent references.
The most important part of this article is the collection and inclusion of the studies in this review/analysis:
1) I see different terminologies such as records, reports, studies, and articles. Please define them in the article as appropriate;
2) I am not entirely sure the data collection was done in this review article. As the figure 3 shows, there were zero related articles in 1990 and 1191, and zero articles in 1996 and 1997. I used the keyword "rabies diagnostics" to search https://pubmed.ncbi.nlm.nih.gov/. When the time was limited 1990-1991 and 1996-1197, respectively, there are more than 100 related publications for each period. I glanced at the hits, and saw lots of articles related to this subject of diagnostics but your figure showed zero articles.
3) when the diagnostic accuracies are compared, probably you need to use one test as the standard. This test has to be the gold standard test, based on my knowledge.
Author Response
This review article targets an important subject: rabies diagnostics. This topic is a great match to this journal, and interesting for the readers. The authors are encouraged to address the comments below.
Answer: We thank the reviewer for your kind and supportive comments.
1.The authors are encouraged to polish the entire manuscript to maintain the accuracy. For example, in the manuscript title, the comprehensive and systemic are redundant, and you need only one of these two words; the writing of "human and canine diagnostics" is very confusing. You can consider modifying the title as: "Evaluating Rabies Test Accuracy: A Systematic Review and Meta-Analysis of Human and Canine Diagnostic Methods."
Answer: We appreciate your input to improve the clarity and accuracy of the manuscript. The title has been revised and changed as per your suggestion: "Evaluating Rabies Test Accuracy: A Systematic Review and Meta-Analysis of Human and Canine Diagnostic Methods."
2.Also in the manuscript abstract, the text reads "Compared to the direct fluorescent antibody test (DFAT), the area under the curve for false positive rates (AUCFPR= 0.887) showed lower accuracy." I do not understand this sentence, and you are comparing the accuracy between which assays.
Answer: Thank you for your valuable feedback. We apologize for any confusion caused by the original wording. The sentence in question compares the diagnostic performance of the DFAT with that of other assays—specifically ELISA, RT-PCR, and RIT—using the area under the curve for false positive rates (AUCFPR) as a measure of accuracy. We have revised the abstract to improve clarity based on your suggestion. We hope this adjustment resolves the ambiguity and provides a clearer understanding of the assays being compared.
3.All the figures need to have a figure legend. For example, for a busy figure 3, the readers can understand this busy figure without a figure legend.
Answer: Thank you for your valuable comment regarding the need for detailed figure legends. We acknowledge that clear legends are essential for improving the interpretability of complex figures like Figure 3. In response, we have revised the legend to provide a more detailed explanation of its components, ensuring that readers can easily understand the figure.
4.As a review article, please include the newest references. I see that there is only one 2024 reference here. Please add more recent references.
Answer: Thank you for your observation. I appreciate your suggestion regarding the inclusion of more recent references. We have reviewed the article and have now incorporated additional references from 2024 to ensure it reflects the latest research in the field.
5.The most important part of this article is the collection and inclusion of the studies in this review/analysis: 1) I see different terminologies such as records, reports, studies, and articles. Please define them in the article as appropriate;
Answer: Thank you for your observation. We appreciate your observation regarding the use of different terminologies. For consistency and clarity, we will use the term 'studies' throughout the article. We will ensure that this is defined appropriately in the revised version.
2) I am not entirely sure the data collection was done in this review article. As the figure 3 shows, there were zero related articles in 1990 and 1191, and zero articles in 1996 and 1997. I used the keyword "rabies diagnostics" to search https://pubmed.ncbi.nlm.nih.gov/. When the time was limited 1990-1991 and 1996-1197, respectively, there are more than 100 related publications for each period. I glanced at the hits, and saw lots of articles related to this subject of diagnostics but your figure showed zero articles.
Answer: Thank you for your observation and for taking the time to carefully review our work. We would like to clarify that, in addition to the keywords "rabies" and "diagnostics," we used the MeSH terms "sensitivity and specificity" to ensure a more precise focus on studies related to rabies diagnostics. This methodological approach might explain the discrepancy between the number of articles identified in our review and the results obtained using a search based solely on keywords. Additionally, as outlined in our methodology, we adhered to the PRISMA guidelines, which involve strict inclusion and exclusion criteria. This resulted in the selection of studies that met specific standards, and for the periods 1990–1991 and 1996–1997, no studies were identified that fulfilled these criteria. It is possible that articles published during those years were not included because they did not meet the predefined criteria for this systematic review and meta-analysis.
3) when the diagnostic accuracies are compared, probably you need to use one test as the standard. This test has to be the gold standard test, based on my knowledge.
Answer: Thank you very much for your valuable comment. You are quite right that a single reference test is generally preferred for diagnostic accuracy comparisons, particularly when a universally accepted reference standard is available. However, in this study, the included investigations used different reference tests tailored to their specific methodologies and objectives, as shown in Tables 2 and 3.
Round 2
Reviewer 3 Report
Comments and Suggestions for Authors
This reviewer appreciates the hard work of the authors in revising the manuscript. Still, I feel like that the methodology and results/conclusion of this article can be significantly approved.
Comments on the Quality of English LanguageThe language is fine, and some improvement might be necessary.